# A Systematic Review of Marketing Practices Used in Online Grocery Shopping: Implications for WIC Online Ordering

**DOI:** 10.3390/nu15020446

**Published:** 2023-01-14

**Authors:** Leslie Hodges, Caitlin M. Lowery, Priyanka Patel, Joleen McInnis, Qi Zhang

**Affiliations:** 1Economic Research Service, U.S. Department of Agriculture, Kansas City, MO 64105, USA; 2Department of Nutrition, Gillings School of Global Public Health, The University of North Carolina at Chapel Hill, Chapel Hill, NC 27599, USA; 3School of Community & Environmental Health, Old Dominion University, Norfolk, VA 23529, USA; 4University Libraries, Old Dominion University, Norfolk, VA 23529, USA

**Keywords:** WIC, online grocery shopping, retail food environment, marketing, nutrition, food access

## Abstract

The Special Supplemental Nutrition Program for Women, Infants, and Children (WIC) plans to allow participants to redeem their food package benefits online, i.e., online ordering. As grocery shopping online has become more common, companies have developed strategies to market food products to customers using online (or mobile) grocery shopping platforms. There is a significant knowledge gap in how these strategies may influence WIC participants who choose to shop for WIC foods online. This review examines the relevant literature to (1) identify food marketing strategies used in online grocery shopping platforms, (2) understand how these strategies influence consumer behavior and consumer diet, and (3) consider the implications for WIC participants. A total of 1862 references were identified from a systematic database search, of which 83 were included for full-text screening and 18 were included for data extraction and evidence synthesis. The included studies provide policymakers and other stakeholders involved in developing WIC online order processes with valuable information about the factors that shape healthy food choices in the online food retail environment. Findings indicate that some marketing interventions, such as nutrition labeling and food swaps, may encourage healthier food choices in the online environment and could potentially be tailored to reinforce WIC messaging about a healthy diet.

## 1. Introduction

The Special Supplemental Nutrition Program for Women, Infants, and Children (WIC) promotes nutrition and health among low-income and racially/ethnically diverse families [1]. WIC provides supplemental nutritious food, nutrition education (including breastfeeding promotion and support), and referrals to health care and other social services to low-income, nutritionally at-risk women, infants, and children up to 5 years of age [2]. Participants purchase the supplementary foods prescribed to them by the WIC program using an electronic benefit transfer (EBT) card at approved grocery retailers. Under current federal regulations, WIC benefits must be redeemed in the presence of a cashier [3]. However, the US Department of Agriculture (USDA) is considering online ordering as an alternative method for WIC participants to redeem their food benefits [4]. WIC online ordering would allow WIC participants to shop like other shoppers and has the potential to increase WIC food package benefit redemptions since some households do not redeem all their WIC food benefits [4]. In a recent survey of WIC participants conducted by the National WIC Association, two out of three WIC participants said they would like to purchase WIC foods by ordering online and using curbside or in-store pickup, and 52.6% reported lack of access to online shopping as a reason for not fully redeeming their WIC food benefits in the prior 6 months [5].

Although online grocery shopping has the potential to improve access to healthy foods, there is a significant knowledge gap in how the online ordering environment, and particularly marketing of products in the online environment, may shape the shopping experiences and food choices of WIC participants. As grocery shopping online has become more common, there has been increased attention to strategies used to market food products to customers who use online (or mobile) grocery shopping platforms. Retailers have developed sophisticated algorithms to make product recommendations as well as to determine the order of products that appear when a customer searches for an item [6]. Retailers frequently use these strategies to promote less healthy products [6]. There has also been increased attention to marketing strategies as a tool for “nudging” shoppers towards healthier products, especially shoppers at risk for poorer health [7]. Whether aimed at boosting sales and increasing market shares or intended to improve diets, these strategies have implications for consumer behavior and ultimately for consumer health and wellbeing; however, a recent review of the healthiness of online supermarkets noted that few studies have investigated the ways that online supermarkets influence the purchasing decisions of customers [8].

To address this knowledge gap, the current study systematically reviews the relevant literature to (1) identify food marketing strategies used in online grocery shopping platforms, (2) understand how these strategies influence consumer purchases, and (3) consider the implications for WIC participants. The first part of the study provides an overview of the online retail environment and the WIC program and discusses how WIC participants may respond to marketing practices in online grocery shopping differently from other consumers. The second part of the study outlines the methodology used for the systematic review and summarizes the findings from the included studies. A comprehensive search of six electronic databases returned 1862 references, of which 83 were selected for full-text screening and 18 for data extraction and evidence synthesis. The selected studies focused on marketing interventions that occurred when customers searched for/discovered food, selected food, and purchased food, as described in the Path to Purchase framework [9], and were further categorized according to the four P’s of price, product, placement, and promotion [10]. Findings suggest that some marketing interventions, such as nutrition labeling and food swaps, may encourage healthier food choices in the online environment. However, because none of the included studies focused specifically on WIC-eligible or WIC-participating populations, more research is needed to assess whether these types of interventions can improve the food choices of WIC participants in line with WIC messaging about a healthy diet.

## 2. The Online Retail Environment

Options for purchasing groceries online have existed for decades [11]. Prior to the coronavirus disease 2019 (COVID-19) pandemic, online grocery purchases accounted for about three percent of total retail grocery sales [11]. However, the market share of online grocery purchases expanded rapidly during the COVID-19 pandemic [12]. Modes of online shopping include websites and, increasingly, smartphone applications. In a mixed-methods study on online grocery shopping in Maryland, 54% of participants who had shopped online previously used a mobile app, whereas 46% used a website [13].

Online grocery shopping has been positively associated with household income, presence of children in the household, and female sex (primary shopper), and negatively associated with age and food assistance program participation [14]. Studies conducted during the early COVID-19 pandemic (June and July 2020) have reported similar associations; respondents who were younger (<40 years old), more educated, higher income, and/or had children in the home were more likely to have shopped for groceries online or to shop online more frequently [12,15].

Barriers to equitable access to online grocery services have been previously identified [16], including the limited availability of online grocery services in rural areas [17,18], fees (e.g., delivery and service fees, gratuity) [13], the disparate costs of products online compared to in-store, and minimum order requirements [19]. Availability of broadband internet may also be a barrier to online grocery services, particularly in rural areas and on tribal lands, where between 22 and 27 percent of Americans living in these areas lack broadband coverage [20].

Some studies conducted prior to the COVID-19 pandemic have reported limited interest in online shopping among low-income populations [21,22]. However, studies among WIC participants, specifically, indicate a high interest in online shopping [23]. In a survey of WIC participants from eleven states and one ITO conducted by the National WIC Association, 65% of respondents said they would like to shop for WIC foods using online ordering with in-store or curbside pick-up, and 36.4 percent said they would like to shop for WIC foods online with home delivery, even if they had to pay an out-of-pocket delivery fee [5].

As grocery shopping online has become more common, there has been increased attention to strategies used to market food products to customers who use online (or mobile) grocery shopping platforms. These may be more often aimed at unhealthy purchases. Moran and colleagues found that candy, sweets, and snacks made up the largest percentage of foods and beverages marketed in the top revenue-generating online grocery retailers in the United States, and that most marketed products were of poor nutritional quality [6]. However, marketing strategies may also be used to promote health and other social values, such as environmental protection. Additionally, some research shows shoppers may be less likely to make unhealthy food purchases when shopping online vs. in store [24,25], despite such marketing practices.

## 3. WIC Program Background & Policy Context

WIC is administered at the federal level by the USDA’s Food and Nutrition Service and serves over six million individuals a month [26]. WIC food packages, a cornerstone of the WIC program, include foods high in nutrients determined to be beneficial for pregnant, breastfeeding, and postpartum women; infants; and children. These foods are not intended to be a household’s primary source of food or serve as general food assistance. WIC food packages accounted for about fifty-eight percent of WIC costs in FY 2021 [27], but only about six percent of WIC household’s total food expenditures, according to the National Academies of Sciences, Engineering, and Medicine [28]. 

Most state WIC agencies use retail food delivery systems to provide program participants with access to the supplemental foods in their food packages. Retail systems allow participants to obtain supplemental food via an Electronic Benefit Transfer (EBT) card, check, or voucher at retail stores authorized by the agency, although EBT is used by most WIC agencies, except a few Indian Tribal Organizations (ITO) and Puerto Rico [29]. WIC State agencies are not required to authorize all qualified stores; however, they must authorize an appropriate number and geographic distribution of stores to ensure adequate participant access. There are approximately 47,000 authorized WIC vendors nationwide [30], including supermarkets, large and small grocery stores, mass merchandisers, convenience stores, gas station food marts, commissaries, and pharmacies.

Under the retail food delivery system, access to WIC foods may be limited for households that do not live near an authorized WIC retailer or who lack adequate transportation to travel to the nearest WIC authorized retailer. A report by USDA’s Economic Research Service using the Food Acquisition and Purchase Survey (FoodAPS) found that although most WIC participants (86 percent) reported using their own vehicle to do their grocery shopping, seven percent report using someone else’s car, and another seven percent reported walking, biking, public transit, shuttle, or another mode of transportation [31]. Additionally, although most WIC participants reported having access to more than one food retailer, they also reported conducting most of their shopping at food retailers further away than the one closest to them, traveling 3.1 miles on average [31]. Some agencies operate direct distribution systems, where participants pick up supplemental food from designated storage facilities operated by the state or local agency, or home delivery systems, where supplemental food is delivered directly to the participant. These types of systems help to meet the needs of clients who have limited access to retail grocery stores, but have proven to be less cost effective than retail food delivery systems [32].

Online grocery shopping has been a growing trend for U.S. consumers since 2009 [33]. It provides a convenient option for consumers to access food and the COVID-19 pandemic significantly increased its adoption [12]. Current federal regulations require that WIC benefits be redeemed in the presence of a cashier [3], which limits online ordering options for WIC participants. During the recent pandemic, some states received waivers for this requirement, allowing for the possibility of online, telephone, and other innovative approaches to redeeming WIC benefits to emerge. However, these waivers only apply until 30 days after the COVID-19 public health emergency, and therefore they do not provide the long-term certainty that most WIC vendors require to invest in the e-commerce technologies needed for online WIC transactions.

To ensure that WIC customers have access to online grocery shopping like shoppers, the Consolidated Appropriations Act of 2021 required the USDA to establish a task force to “study measures to streamline the redemption of supplemental foods benefits that promote convenience, safety, and equitable access to supplemental foods” [34]. After evaluating alternative methods of WIC redemption, the task force strongly recommended developing rules that allow modern and intelligent ordering and purchasing methods consistent with existing commercial models [34]. Following the task force’s recommendation, the USDA announced their intention to revise existing regulations to allow WIC participants to redeem WIC benefits online. USDA has also partnered with the Gretchen Swanson Center for Nutrition to begin piloting WIC online ordering projects in some states [35].

State agencies have also initiated their own online ordering pilots using a “click and collect” approach. Under this model, WIC participants use a grocery shopping application to order their WIC foods but pick them up at the store or curbside, so that they can redeem their benefits in the presence of a cashier [36,37]. Analyzing the shopping patterns of WIC participants in Oklahoma who had access to a “click and collect” online ordering option, one study found that 40 percent of participants who used the online shopping option did so only once [36]. Another study assessed a similar approach to WIC benefit redemption online in Tennessee [37]. Although all participants in the pilot were able to successfully order benefits online and pick them up in person, several issues emerged, including difficulty identifying WIC items on the grocery store website and having to look through long lists after searching for an item to identify a WIC-approved product [37].

## 4. Online Marketing Considerations for WIC Participants

Due to the design of the WIC program, marketing strategies used in online grocery retail settings could influence WIC customers differently than other customers. First, WIC benefits are for the purchase of a specific quantity of a given item rather than a certain dollar amount. Except for fruit and vegetable purchases, for which WIC participants are provided a fixed dollar amount, we do not expect WIC participants to be influenced by the pricing of items they are purchasing with their WIC food benefits. We would, however, expect WIC customers to be sensitive to prices of foods that they purchase for their households with other resources, such as SNAP or personal income. We also expect that when shopping for WIC food items, WIC participants are typically also shopping for non-WIC food items, since WIC accounts for only about six percent of households’ food budgets [28].

Marketing strategies used in online grocery retail settings could also influence WIC customers differently than other customers because, under current federal regulations, retailers cannot offer promotions that are specific to food assistance program participants [3]. However, retailers can offer promotions that they offer to all other non-WIC shoppers and promotions that are offered to customers with specific characteristics, such as promotions to shoppers with infants or young children. Additionally, WIC participants can take advantage of promotions such as buy one, get one free (BOGO) without having the second (free) food items subtracted from their WIC food benefits [38]. They can also buy larger sizes of items if a manufacturer is offering the larger size at the same price, i.e., a quantity discount [38].

Marketing interventions could also play a role in WIC participants’ abilities to identify WIC-eligible foods. Product promotions, such as in-app advertisements, and product placement, such as search-result order, that focus on branded items could make it more difficult to identify WIC-eligible foods. Some state WIC agencies require that participants purchase the least-expensive brand or store brand of a food item [39]. However, product promotions and placement could also make it easier to identify WIC-eligible foods by labeling products as WIC-approved or adjusting search order to ensure that WIC-approved items appear first in search results. Prior research has indicated that clearer labeling of WIC-approved items creates a better shopping experience for WIC participants and may support WIC benefit redemption and continued participation in the program [40].

Finally, marketing interventions could influence the nutritional quality of non-WIC food purchases. Marketing practices focused on sales of less healthy foods, such as candy, sweets, and snacks make up the largest percentage of foods and beverages marketed by online grocery retailers in the United States [6]. These types of practices could influence WIC participants to use their non-WIC resources for less healthy purchases. On the other hand, marketing strategies focused on sales of more healthy foods could reinforce WIC messaging about healthful food choices and improve the healthfulness of WIC participants’ non-WIC food purchases. Recent research suggests that WIC participants’ non-WIC food purchases are not as healthful as their WIC food purchases. Fang et al. compared the nutritional quality of food purchases made by WIC participants who did and did not use their WIC benefits during a shopping trip [41]. They found that nutritional quality of food purchases was higher among those WIC participants who redeemed their WIC benefits during a shopping trip.

## 5. Materials & Methods

This systematic review was registered with Prospective Register for Systematic Reviews CRD 42022339637 (https://www.crd.york.ac.uk/prospero/display_record.php?ID=CRD42022339637, accessed on 8 December 2022).

### 5.1. Data Sources

Six electronic databases were searched for applicable articles in the public health, behavioral sciences, and business literature: PubMed, EMBASE, Business Source Complete, PsycINFO (via EBSCOhost), EconLit (via EBSCOhost), and ABI/Inform. The search took place in May and June of 2022 and aimed to identify studies focused on retail marketing strategies used in the online grocery shopping environment.

### 5.2. Search Strategy

A search strategy using Boolean operators and subject heading terms was developed in consultation with a Health & Life Sciences Librarian at Old Dominion University and in consultation with six experts in areas of retail marketing, online grocery retail, e-commerce, digital literacy, consumer protection, public policy, and public health. Experts were identified through membership in the Healthy Eating Research and Nutrition & Obesity Policy Research & Evaluation Network’s WIC Learning Collaborative or through known contacts of members of the WIC Learning Collaborative. After an initial consultation regarding key terms and constructs to consider in the literature search, the experts received the draft search strategy and were asked for feedback. Search terms included parent, single parent, child, children, infant, consumer, shopper; supermarket, food retailer, grocery shopping, food purchase, food shopping, food choice; online, internet, virtual, web, web-based, e-commerce, app, cyber, digital; marketing, advertising, promotion, retail strategies, retail marketing, online retail, retail analytics, targeted advertising, targeting marketing, data mining, web analytics, and consumer analytics. The complete search strategy of the electronic databases is described in Appendix A, and the complete search strategy of the gray literature (conducted via Advanced Google) is described in Appendix B.

### 5.3. Eligibility Criteria

The review included studies in English published on or after 1 January 2015. Conversations with project consultants indicated we would find the most relevant studies by limiting the search to studies published after 2014. Moreover, a related study identified only three studies on the impact of online grocery retail on consumer purchases and dietary patterns, all of which were published after 2014 [8].

Studies could be interventional, observational, or qualitative and implemented in real-world or lab settings. Studies were peer-reviewed or identified through a systematic search of grey literature. Systematic, scoping, or narrative reviews; conference or dissertation abstracts; news articles, and other general information articles were excluded. Studies that focused on retail marketing strategies such as promotions, discounts, and limited deals used as consumers shopped for groceries online were included. Studies focused on marketing via social media platforms, television, or content streaming applications were excluded. Included studies were not limited to any specific group of consumers. However, efforts were made to ensure that the search strategy captured studies that focused on populations like those eligible for participation in WIC, such as parents with young children living in households with low incomes. Study screening was completed using Covidence, following the methods outlined in the Preferred Reporting Items for Systematic Reviews and Meta-Analyses (PRISMA) guidelines. The population, intervention/exposure, comparison, outcome, and study design (PI(E)COS) criteria, which determined whether studies were eligible for inclusion, are described in Table 1.

### 5.4. Data Extraction

Two researchers (C.L. and P.P.) completed the extraction of the title, abstract, and full text using Covidence with a random agreement of 89% for title and abstract screening. The two analyzed each study independently and then met to resolve disagreements. Two additional researchers (L.H. and Q.Z.) were consulted when disagreements persisted. JBI’s critical appraisal tools were used to assess the methodological quality of included papers [42]. JBI has tools (checklists/guidelines) for most study types. These tools allowed the research team to determine the extent to which bias was addressed in the included studies.

### 5.5. Evidence Synthesis

The Path to Purchase framework developed by Khandpur et al. guided evidence synthesis [43]. Extraction was informed by marketing interventions under “personalized marketing by the retailer”, although for the purposes of the current study the interventions could be researcher or retailer initiated. These types of interventions occurred during three phases of online shopping (customer searches for/discovers food, selects food, and purchases food) described in the framework. The marketing interventions were further categorized by the four P’s of price, product, placement, and promotion, a common typology in marketing research [10]. Additional data extraction included study type, methodology, and size and characteristics of the study sample. The limited research on the topic and the variety of study designs and methods among the included studies precluded meta-analysis. (Table 2)

## 6. Results

The search criteria returned a total of 1862 references, which dropped to 1590 after the removal of duplicates (Figure 1). After title and abstract screening, 83 articles from the database search were included for full-text screening. Of these articles, 18 met the inclusion criteria and were included for data extraction and evidence synthesis, see Table 2. Of the eighteen articles, nine studies were randomized controlled trials [46,49,50,51,52,54,55,56,59]. Another seven [44,45,47,48,53,57,60] were observational studies that primarily used quasi-experimental methodologies such as analyzing retail scanner data to follow the behavior of shoppers over time. Two studies included both observational and experimental methods [24,58]. We also conducted a grey literature search to identify relevant studies that may not have been peer reviewed or identified in search of academic databases. Our grey literature search did not return any additional studies that met the inclusion criteria. The details of our grey literature search are reported in Appendix B.

### 6.1. Study Characteristics

The included studies listed in Table 2 were all published between 2015 and 2022. Studies were conducted in Belgium [49], Denmark [56], Iceland [58], Iran [59], the Netherlands [52], Spain [44], the United Kingdom [45,46,48,50,54,55], the United States [53,57,60], and multiple or unspecified European countries [24,47,51].

Almost all studies used data collected from online grocery store settings. In several studies [24,49,50,51,52,54,55,56], the online grocery store was “mock” or “virtual”; this was largely the case when studies were randomized controlled trials. In several other studies [24,45,47,48,53,60], data were also collected from offline brick-and-mortar stores to compare purchase behaviors and nutritional quality of food choices across channels. Sigurdsson and colleagues used survey data collected from consumers about their online shopping experiences to draw conclusions about the role of marketing practices in food purchases [58].

Most studies targeted adult populations or households (broadly defined) in the countries in which they were conducted. For instance, Forwood et al. recruited a nationally representative sample of adult participants in the U.K. for their study [50], whereas De Bauw et al. targeted “food decision makers” in Belgian households [49]. Studies using retail scanner data or household loyalty card data sometimes did not explicitly state that they restricted their analyses to adult shoppers or certain types of households but often restricted to households who made certain numbers of online or in-store purchases [44,45,46,47]. Several studies conducted subgroup analyses, but no studies we identified were specifically targeted at the WIC population or populations that could be inferred to be WIC or WIC-eligible populations, such as mothers of young children living in households with low incomes.

### 6.2. Interventions and Outcomes

The identified studies examined marketing inventions (manipulations of the shopping environment) that WIC customers may experience when purchasing WIC foods and other groceries online. They also examined marketing interventions that have been found to promote healthy food choices that could potentially be incorporated into the development of WIC online shopping platforms. We organized the studies into four categories of marketing interventions: price, product, placement, and promotion. Studies in the price category include interventions such as increasing or decreasing the price of food items at different times or for different customers (dynamic pricing) as well as adjusting pricing of items relative to other online retailers or relative to the same retailer in the offline channel. Studies in the product category manipulate the shopping environment by using prior purchase information to make recommendations, by providing information about products (through labeling and advertisements), and offering swaps of products with certain characteristics as alternatives to the shopper’s initial choice. Studies in the placement category include interventions that manipulate the layout of the online grocery store environment. Studies in the promotion category focused on interventions that offered deals, such as buy one, get one free (BOGO). Many studies combined multiple approaches. All studies focused on consumer behavior (food purchases).

### 6.3. Price

Campo and Breugelmans used retail scanner data from a larger European retailer that offered online and instore shopping [47]. They were interested in how price manipulations as well as other aspects of the marketing mix (assortment, promotion, and in-store stimuli) influenced shopping behaviors within and across online and offline channels. They found that if the same retailer offered different (higher) prices online than offline, this deterred those who were already skeptical about online shopping from using the online channel for grocery purchases [47]. However, shoppers who had more online grocery shopping experience were not deterred by higher prices online than offline, perhaps because they valued other benefits of online grocery shopping, nor were new fans of online grocery shopping, who were more influenced by product assortment online (i.e., availability of a wide range of products).

Richards et al. examined how offering local foods affects online grocery store retail pricing [57]. Although they noted that the availability of local foods at grocery stores and supermarket chains has largely been driven by consumer demand for locally produced and sourced foods, they found that offering more local foods allowed online grocery retailers to raise prices of complementary non-local foods, as consumers prefer to buy local and non-local foods during a single shopping trip, ultimately leading to higher profit margins and sales volumes for retailers.

Two additional papers, Sigurdsson et al. [58] and Vahdani and Sazar [59], focused on pricing in conjunction with customer reviews and found them to be a powerful marketing tool that can be used to manipulate prices and purchasing behaviors in the online shopping environment. In Sigurdsson et al., respondents to an online survey were asked to imagine that they were shopping for salmon online [58]. Respondents were shown four options where country of origin, procurement method (wild caught vs. farmed), price, signage (product marked as “store’s choice” or “top seller”), purchase state (fresh vs. frozen), delivery method (same day vs. next day), and product rating (3 to 5 stars) varied. Analysis of their responses indicated that product rating (number of stars) mattered the most for purchasing decisions, whereas price had a more moderate influence (ranking below product rating, procurement method, and country of origin). In a secondary analysis of data from an online retailer in Iran, Vahdani and Sazar examined the role of reviewer comments and pricing in customers’ purchases of perishable products, specifically, pasta salad [59]. They found that a high rating generally offset customers’ disinclination to buy products that were approaching their expiration date. They propose a dynamic pricing model allowing retailers to reduce the typical discount offered on products nearing their sell-by date for perishable products with high ratings, potentially reducing food inventory waste.

### 6.4. Product

Retailers have already developed sophisticated algorithms to personalize product recommendations for online customers [61]. Customers shopping for groceries online are likely to encounter product recommendations based on these algorithms when initiating shopping in the application, searching for or after selecting items, and/or at checkout. Among our review studies, Lee and colleagues were interested in identifying ways to improve on the abilities of existing recommender models to accurately predicted future food purchases [53]. They found that incorporating information about purchase order of items not only improved the accuracy of existing model’s predictions of future purchases, but it also increased the diversity of recommended items, which was a finding likely to interest retailers and manufacturers looking to increase sales and market share. From the shopper perspective, the use of this type of algorithm may expose them to a broader range of products. For example, recommender models may be used to expose shoppers to products through advertisement banners and recipes. In Bunten et al., shoppers who were exposed to advertisement banners and recipes that used healthier products purchased more of the promoted healthier items [46].

Swaps are another type of recommendation system or algorithm. Studies that manipulate the food environment by offering swaps typically suggest alternatives that align with a public health or social goal, such as items that are healthier or more environmentally friendly. In Forwood et al., participants were asked to shop for 12 items in an experimental online supermarket [50]. Intervention groups received an average of four prompts to swap one of the twelve food items for another item, with a median swap acceptance of one [50]. Swaps were offered at product selection and at checkout and were more likely to be accepted at product selection. Women, a key demographic in WIC, were more likely to accept swaps. Swaps were intended to promote healthier items, and there was a reduction in energy density for each swap accepted (typically resulting from a lowering in saturated fat). However, because the number of accepted swaps was low (a median of one), this typically did not result in an overall reduction in the energy density of food purchases compared to those who were not offered swaps.

In addition to recommender models and swaps, our review studies overwhelmingly demonstrate the influence of information–especially information provided through labeling–on food choices in online settings. Much of the research focuses on the Nutri-score nutrition label which was first adopted in France and is now used in several other European countries, such as Belgium, Spain, Germany, Switzerland, the Netherlands, and Luxembourg [62]. Among our review studies, Zou and Liu [60] and Fuchs et al. [51] found that displaying a nutrition label, such as the Nutri-score nutrition label, on food products during the shopping process led to healthier shopping behaviors. In Fuchs et al., a web-browser extension was developed and used to display the Nutri-score as participants shopped on the platform of an online grocery retailer [51]. The researchers purposely developed the web extension to make nutrition information broadly available to online shoppers rather than conditional on adoption or implementation by a specific retailer.

As is increasingly common in retail marketing studies interventions were often combined [7]. For example, Jansen and colleagues combined the Nutri-score label with another intervention, swap offers [52]; De Bauw and colleagues combined product scores (Nutri- and Eco-score labels) with product recommendation agents [49]; Marty and colleagues combined nutrition labeling with product assortment [54]. All studies found that nutrition labeling resulted in healthier food choices. In subgroup analyses conducted by Marty and colleagues, labeling was mainly impactful for those with existing health motives [54]. Although this indicates that nutrition labeling may not have uniform behavioral effects, it also indicates that food labeling could be particularly effective for WIC participants for whom existing health motives are a likely factor in their decision to participate in the program.

### 6.5. Placement

How foods are presented to shoppers in online shopping environments may also influence food choices. Huyghe and colleagues considered differences in vice purchases (unhealthy food purchases) between customers shopping in a traditional online store (where participants select a category and view products within category) and an “uncategorized” online store (where participants viewed products as they appeared on physical shelves) [24]. They found no differences in vice purchases in the two settings. However, they did observe that vice purchases were less common in both online settings than they were in the offline settings, suggesting that online shopping in general may promote healthier food choices (or fewer unhealthier food choices/vice purchases).

From Panzone et al. it may also be possible to infer that online shopping environments that organize food by healthfulness could facilitate healthier food purchases [55]. The study did not focus on the healthiness of foods, but instead organized grocery products into aisles based on their carbon footprint, with highest carbon footprint foods in one aisle, medium in another, and low in a third. This reduced the amount of work that shoppers had to do to determine an item’s carbon footprint. They observed a significant decreased in the carbon footprint of items purchased by consumers exposed to this organization.

### 6.6. Promotions

Offering deals for the purchases of certain products or a certain quantity of a product is another way that the online shopping environment can be manipulated. However, findings from our review indicate that customers are more sensitive to promotions when shopping offline (in traditional brick-and-mortar grocery stores) than they are when shopping online. Arce-Urriza, Cebollada, and Tarira conducted secondary analysis of scanner data to examine how shoppers responded to promotions for orange juice [44]. They found no effect of online promotions on online purchases of orange juice and a statistically significant effect of offline promotions on offline purchases of orange juice, perhaps due to heterogenous effects of promotions on different types of shoppers. For example, Campo and Breugelmans found that promotions were particularly important for stimulating purchases among new online shoppers [47], and Breugelmans and Campo found that promotions offered in online settings tended to stimulate online purchases and suppress offline purchases [45]. In essence, shoppers offered promotions when shopping online were motivated to continue to shop online.

The review studies also provide information about the types of promotions offered to online shoppers and how they influence food purchases. Peschel examined the number of items customers purchased when exposed to two common promotions: abundance signaling (such as 10 items for $10) and scarcity signaling (such as going fast or limited supply) [56]. They found that both types of promotions led to purchases of more items (vs. no promotion), with abundance signaling leading to more purchases than scarcity signaling. In additional analyses, they considered differences by interacting customers’ body weight and income. In the scarcity signaling scenario, fewer units were purchased by those who had higher incomes and were overweight compared to those who had lower incomes and were overweight. Similar numbers of items were purchased by low- and high-income individuals who had normal weights. In the abundance signaling scenarios, weight status was positively associated with buying more items in each category and this relationship did not differ by income. On the upside, these findings suggest that in the online environment, shoppers who are at higher risk of poor health due to overweight/obesity may be more inclined to purchase healthier products when exposed to promotions that indicate that the product is scarce. On the downside, the same is likely to be true if scarcity signaling is applied to unhealthy items.

## 7. Discussion

This project systematically reviewed the scholarly literature to identify studies on how food marketing strategies used in online grocery shopping platforms influence consumer food preferences and food purchases and to consider the implications of the findings from these studies for WIC online ordering. In doing so, our paper contributes to the emerging literature on the characteristics of online shopping environments [6,8] and the marketing practices of online retailers, which include sophisticated algorithms that make product recommendations; promotions that indicate that items are scarce (going fast) or abundant (buy many at a lower price); and advertisements and promotions designed to increase related-item purchases.

Findings from the literature that focused on pricing and product assortment point to characteristics of the online shopping environment that may encourage or discourage WIC participants from online grocery shopping. Although we do not expect prices of WIC food items to shape the shopping decisions of WIC participants, WIC participants may avoid online shopping if they perceive non-WIC food items are more expensive online compared to offline. They may also make decisions about online versus offline shopping depending on product assortment (i.e., whether the same products are available online as offline) and the types of promotions they are offered in online and offline channels [45,47,48].

Findings from the literature that focused on pricing also suggest a need for future research to examine how WIC participants respond to pricing of fresh produce in online settings. Low-income families have expressed concerns about purchasing fresh produce online since they cannot verify the quality of items by touching or examining them [13]. In lieu of being able to lay eyes and hands on a product, online shoppers may obtain information about a product’s quality from product ratings and reviews left by other customers. However, two papers in our review study suggest that retailers can use positive product ratings and reviews to charge higher prices for perishable products online [58,59], which could result in WIC participants (and other shoppers) paying more for the same amount of fresh produce online compared to offline. It could also influence shoppers’ decisions about purchasing frozen or canned produce rather than fresh produce online. In an analysis of a survey sample of low-income shoppers, Trude and colleagues found that families who shopped online were less likely to buy fresh fruits, fresh vegetables, and meats or seafood [13]. 

Findings from our study complement studies on the effectiveness of interventions that manipulate “the four P’s” to promote healthy food purchases in offline grocery retail settings [7]. Although the marketing interventions in the studies we identified did not have to be targeted towards the promotion of healthier food purchases, many were. Their findings indicate the potential for future studies to consider how the online shopping environment could be tailored to reinforce WIC nutrition education regarding a healthy diet.

For example, one study found that foods such as pasta, rice, and noodles had the greatest acceptance of swaps and that women in the study were more likely to accept food swaps than men [50]. Future studies could examine whether interventions using food swaps could be used to encourage WIC participants to purchase healthier non-WIC foods. As noted earlier, there is room for improvement in this area, as the food purchases of WIC participants are less healthy when they do not use WIC benefits during a shopping trip [41].

It is also interesting to consider the potential for WIC to “train” recommender models towards healthier foods. For split-tender transactions, WIC participants must first use WIC to pay for items they are purchasing and then pay for remaining items with other means, such as SNAP or personal income, making it likely that WIC items are purchased first. If recommendation models use purchasing order as a sign of preference for food items (with those purchased first being the most preferred) [53], this may lead these algorithms to make future recommendations for foods that are similar in healthfulness to those included in the WIC food packages.

Providing recipes and ingredient lists featuring WIC foods at a point that allows the items to be added directly to one’s online shopping cart may facilitate the redemption of benefits. The WIC shopper app already provides recipes that incorporate WIC foods to participants to help promote purchases and consumption of WIC foods, but evidence of WIC participant interest in this feature is mixed [63]. However, the WIC shopper app is not an online grocery shopping platform, but rather a way for WIC participants to track what benefits they have available and identify food items that qualify for WIC. Future research could examine whether WIC participants would be more interested in a similar type of feature that was embedded directly into their online shopping application and whether this increased redemption of WIC foods.

Nutrition labeling stood out as a promising marketing intervention for promoting healthy food choices. Not all studies have found impacts related to nutrition labeling impacts purchases. For example, a study conducted in 2011 Australia found that traffic-light nutrition information, a labeling approach that uses color coded information to display a product’s levels of fat, sugar, and sodium, did not influence online grocery sales [64]. However, the more recent studies included in the current review that focused on nutrition labeling found that it led to healthier food purchases, which is an especially salient finding given that several recent studies have pointed to online grocery stores’ failures to provide even required nutritional information in the United States [8,65,66] and elsewhere [67]. In one study, a web-browser extension was developed and used to display the Nutri-Score as participants shopped online [51]. The extension was purposely created to operate in any online grocery retail setting so long as it was being access via the web-browser. Similar web-browser extensions could be made available to WIC participants to use when shopping online to help them maximize the healthfulness of the food items that they select within a food package item category, such as dry cereal, as well as help them to maximize the healthfulness of additional non-WIC food items that they are purchasing for their households.

### Limitations

The study findings should be interpreted in the context of several limitations. First, no studies that we identified focused specifically on the WIC population or subpopulations that could be inferred to be WIC-eligible or WIC-participating. This suggests a need for future research to replicate these interventions with policy-relevant subpopulations, such as families with young children and pregnant women.

Next, our review focused on research where the intervention was limited to retail marketing strategies used in online grocery retail settings; however, this does not fully capture all the possible ways in which retailers and other parties/stakeholders may influence food choices of consumers who grocery shop online. For example, our review does not include studies of interventions that were not explicitly marketing strategies (price, product, placement, promotion), even if the intervention took place in an online grocery retail setting and measured outcomes such as food purchases and food preferences. One excluded study focused on a public health intervention where customers received information about the benefits of dietary fiber. The information provided resulted in an increase in higher fiber purchases [68]. The study provides important information regarding the value of interventions that provide information about the dietary quality of foods to customers. Although outside the scope of the current study, which defined marketing practices as the intervention, this type of study certainly has implications for WIC online orders when considering approaches that could reinforce nutrition education provided through the WIC program. Our review also does not include studies of marketing interventions that measured non-food purchase behaviors such as customer loyalty and patronage (shopping with the same retailer again or intention to shop with the same retailer again), which is often the goal of marketing practices used by retailers.

Though not necessarily a limitation, our review did not consider studies focused on food marketing on social media platforms, food marketing in content streaming apps (e.g., Hulu), and food marketing on television. A recent synthesis of reviews of the association between exposure to unhealthy food advertising on TV and online and childhood obesity found a positive dose-response relationship, with stronger associations for children aged 3–12 years, low-income groups, and racially and ethnically diverse populations [69]. Additionally, we did not focus on whether the online grocery store settings offered home delivery, curbside pickup, or both. If studies only include home delivery (i.e., some earlier European studies), the sample may be biased towards populations in particular areas (where delivery service is available) and those with higher income, who are comfortable with paying delivery fees and potentially higher prices for online products, and who are not concerned about minimum order requirements, which would not necessarily apply to curbside pickup. Studies that examine grocery delivery may be relevant to WIC participants, as grocery delivery services are often used by mothers and other caretakers of young children [14,15]. Although not covered by our study, these other domains are important areas for future research on how retailers and other parties/stakeholders may influence food choices of consumers.

## 8. Conclusions

WIC is a federal program that provides nutritious food and nutrition education to low-income, nutritionally at-risk women, infants, and children up to 5 years of age. Participants use an electronic benefit transfer (EBT) card to purchase supplementary foods at approved retailers, with benefits typically required to be redeemed in person in the presence of a cashier. As USDA is considering allowing WIC participants to redeem their benefits online, this paper aimed to address a significant knowledge gap in understanding how food marketing strategies used in online grocery shopping platforms influence consumer purchases and consider the implications for WIC participants. The findings from 18 studies that met the inclusion criteria suggest that characteristics of the online shopping environment, such as pricing and product assortment, may influence decisions regarding online versus offline shopping, which could inform future studies interested in identifying when WIC participants might choose an online option. Additionally, some retailers may use positive customer ratings and reviews to charge higher prices for perishable products online, which could limit the ability of WIC participants to maximize the value of their cash-value benefits when purchasing fresh fruits and vegetables online. The review also identified ways in which the online shopping environment could potentially be tailored to reinforce WIC nutrition education and encourage healthier food purchases. Future studies could examine the use of food swaps and the potential for WIC food purchases to “train” recommender models towards healthier foods as strategies to promote healthier non-WIC food purchases among WIC participants.

## Figures and Tables

**Figure 1 nutrients-15-00446-f001:**
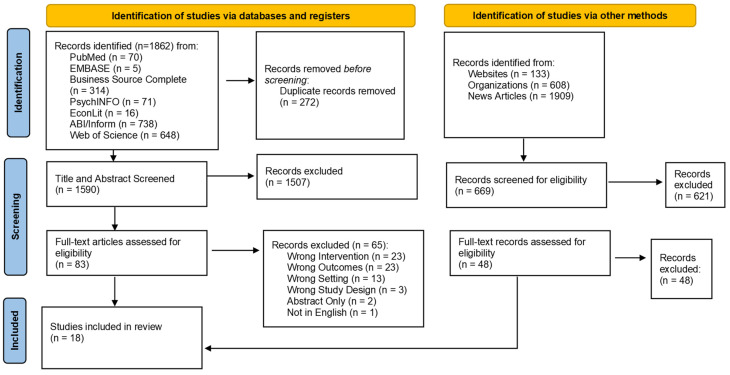
Flow diagram of studies included in the systematic review. Notes. Adapted from Page MJ, McKenzie JE, Bossuyt PM, Boutron I, Hoffmann TC, Mulrow CD, et al., “PRISMA 2020 flow diagram for new systematic reviews which included searches of databases, registers, and other sources”. The PRISMA 2020 statement: an updated guideline for reporting systematic reviews. BMJ, 372. doi:10.1136/bmj.n71.

**Table 1 nutrients-15-00446-t001:** PI(E)COS criteria for inclusion and exclusion of studies.

Parameter	Inclusion Criteria	Exclusion Criteria
General	Published in an English peer-reviewed publication or publicly available government or non-governmental report	
Study design	Quantitative and qualitative, could be quasi-experimental, experimental, longitudinal, cross-sectional, observational.	Conference proceedings or abstracts, dissertations or theses, news articles
Population	Consumers	
Intervention/exposure	Marketing strategies (product suggestions, promotions, price, etc.)	
Setting	Online grocery shopping platform	Other online settings such as social media platforms (e.g., Facebook), non-grocery e-commerce sites (e.g., clothing retailers), brand sites (e.g., Coca-Cola, Pepsi), and content streaming sites (e.g., YouTube, Hulu); physical grocery stores; television; postal mail
Outcomes	Food purchases as well as outcomes such as nutrient and diet quality of foods purchased. Food preferences indicated by consumers via a survey or other method of data collection.	

**Table 2 nutrients-15-00446-t002:** Summary of Included Studies.

References	Study Design	Setting	Population	Intervention	Outcome	Quality Assessment Tool Used
Arce-Urriza, M., Cebollada, J., & Tarira, M. (2017). The effect of price promotions on consumer shopping behavior across online and offline channels: Differences between frequent and non-frequent shoppers. [44]	OBS	Multi-channel (online and offline)	Shoppers that were loyalty card members at a Spanish grocery chain, who purchased orange juice at least twice during the study period, and were multi-channel shoppers	Brand-specific price promotions on orange juice	Purchase of 1-L of orange juice across online and offline channels	Quasi-experimental
Breugelmans, E., & Campo, K. (2016). Cross-Channel Effects of Price Promotions: An Empirical Analysis of the Multi-Channel Grocery Retail Sector. [45]	OBS	Multi-channel	Shoppers who purchase milk or cereal and shopped at Tesco in the UK	Price promotions on milk and cereal	Purchase of milk and cereal	Quasi-experimental
Bunten, A., Shute, B., Golding, S. E., Charlton, C., et al. (2022). Encouraging healthier grocery purchases online: A randomised controlled trial and lessons learned. [46]	EXP	Multi-channel	Shoppers with loyalty card at large chain retailer (Sainsbury’s)	Advertisement banners and ingredient lists with healthier versions of products and recipes	(1) Primary: purchases of healthier products; (2) secondary: banner clicks, purchases of standard products, overall purchases, and energy (kcal) purchased	RCT
Campo, K., & Breugelmans, E. (2015). Buying Groceries in Brick and Click Stores: Category Allocation Decisions and the Moderating Effect of Online Buying Experience. [47]	OBS	Multi-channel	Shoppers that were loyalty card members with at least two online and two offline purchases at a single multichannel retailer, and at least two purchases in a given category	Marketing-mix	Share of food category spending purchased online	Quasi-experimental
Campo, K., Lamey, L., Breugelmans, E., & Melis, K. (2021). Going Online for Groceries: Drivers of Category-Level Share of Wallet Expansion. [48]	OBS	Multi-channel	Shoppers from a household panel who started online grocery shopping during the study period at one of four major multi-channel retailers and purchased items in a food category (a) before and after they began online shopping and (b) purchased the category at more than one chain (before or after they started online shopping) of the ten chains (four multi-channel, six single-channel chains) included in the study	Marketing-mix	Share of food category spending purchased online	Quasi-experimental
De Bauw, M., De La Revilla, L. S., Poppe, V., Matthys, C., & Vranken, L. (2022). Digital nudges to stimulate healthy and pro-environmental food choices in E-groceries. [49]	EXP	Online (mock)	Household food decision makers (representative sample of Dutch-speaking Flemish adults)	Product recommendation agents, Nutri- and Eco- score labeling, personalized social norm messages	Nutritional quality and environmental impact of purchases	RCT
Forwood, S. E., Ahem, A. L., Marteau, T. M., & Jebb, S. A. (2015). Offering within-category food swaps to reduce energy density of food purchases: A study using an experimental online supermarket. [50]	EXP	Online (mock)	Nationally representative sample of adults in UK who did more than half of household’s food shopping	Food swaps with a “consented” introductory message or an “imposed” introductory message	Energy density of shopping basket and proportion of swaps accepted	RCT
Fuchs, K. L., Lian, J., Michels, L., Mayer, S., Toniato, E., & Tiefenbeck, V. (2022). Effects of Digital Food Labels on Healthy Food Choices in Online Grocery Shopping. [51]	EXP	Online (mock)	College students	Nutri-score labeling via a web browser extension that displayed a label next to each product	Purchases of healthier foods	RCT
Huyghe, E., Verstraeten, J., Geuens, M., & van Kerckhove, A. (2017). Clicks as a Healthy Alternative to Bricks: How Online Grocery Shopping Reduces Vice Purchases. * [24]	OBS/EXP	Multi-channel	College students	Traditional online mock store platform (searchable by category), uncategorized online mock store platform (click images organized like offline store), offline mock-store	Vice purchases	Quasi-experimental/RCT
Jansen, L., van Kleef, E., & Van Loo, E. J. (2021). The use of food swaps to encourage healthier online food choices: A randomized controlled trial. [52]	EXP	Online (mock)	Adults in the Netherlands	Swap offer, Nutri-score labeling, descriptive norm messaging	Nutrient profile score of food choices	RCT
Lee, H. I., Choi, I. Y., Moon, H. S., & Kim, J. K. (2020). A Multi-Period Product Recommender System in Online Food Market based on Recurrent Neural Networks. [53]	OBS	Online	Customers of online fresh food delivery service company in the United States	Product recommendation system	Accuracy of predicting shoppers’ future purchases	Quasi-experimental
Marty, L., Cook, B., Piernas, C., Jebb, S. A., & Robinson, E. (2020). Effects of Labelling and Increasing the Proportion of Lower-Energy Density Products on Online Food Shopping: A Randomised Control Trial in High- and Low-Socioeconomic Position Participants. [54]	EXP	Online (mock)	Adults with access to computer and internet in UK who were main shopper for their household	Labeling and assortment of lower energy density products	Energy density of items in shopping basket	RCT
Panzone, L. A., Ulph, A., Hilton, D., Gortemaker, I., & Tajudeen, I. A. (2021). Sustainable by Design: Choice Architecture and the Carbon Footprint of Grocery Shopping. [55]	EXP	Online (mock)	College students	Choice architecture, moral goal priming, different tax-scenarios	Carbon footprint of basket of goods chosen by the consumer	RCT
Peschel, A. O. (2021). Scarcity signaling in sales promotion: An evolutionary perspective of food choice and weight status. [56]	EXP	Online (mock)	Adults	Scarcity siganling and abundance signaling across type of prodcut (storable/parishable, healty/unhealthy)	Number of units chosen within the different product categories	RCT
Richards, T. J., Hamilton, S. F., Gomez, M., & Rabinovich, E. (2017). Retail intermediation and local foods. [57]	OBS	Online	Shoppers at Relay Foods, an online retailer in Virginia	Assortment of local products	Total sales	Quasi-experimental
Sigurdsson, V., Larsen, N. M., Alemu, M. H., Gallogly, J. K., Menon, R. G. V., & Fagerstrøm, A. (2020). Assisting sustainable food consumption: The effects of quality signals stemming from consumers and stores in online and physical grocery retailing. ** [58]	OBS/EXP	Multi-channel	Facebook group; adult shoppers	Quality signals (customer ratings and store recommendations)	Purchase of fresh fish	Quasi-experimental/RCT
Vahdani, M., & Sazvar, Z. (2022). Coordinated inventory control and pricing policies for online retailers with perishable products in the presence of social learning. [59]	EXP	Online	Shoppers at online supermarket in Iran	Expiration date based pricing and quality signals (customer ratings via online review system)	Product inventory	RCT
Zou, P., & Liu, J. W. (2019). How nutrition information influences online food sales. [60]	OBS	Online	Customers shopping at online retailers in China via the shopping platform Taobao	Labeling and quality signals	Food sales (six products)	Quasi-experimental

* Only the experimental portion of the study met our inclusion criteria. ** Only the observational portion of the study met our inclusion criteria.

## Data Availability

Not applicable.

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
