# Peer review of "A Systematic Review of Marketing Practices Used in Online Grocery Shopping: Implications for WIC Online Ordering"

_nutrients, 2023, doi:10.3390/nu15020446_

Round 1

Reviewer 1 Report

Manuscript: Systematic Review of Marketing Practices Used in Online Grocery Shopping: Implications for WIC Online Ordering

The authors have submitted a manuscript on an original and little-discussed topic of great importance. This work is well written and its structure is precise and direct. I believe that it has the potential to be proceed since it succinctly collects the consumption patterns of different shoppers and online food purchases. Specific comments follow:

1. Please, add references.

2. Please, review the sum of all records identified. Based on my calculations it should be “1,862” the figure reads “Records identified n=1,857”.

3. Add the limitations of the study at the end of the discussion section.

Author Response

Thank you for reviewing our paper and providing suggestions for revisions. These suggestions have helped us to strengthen the paper.

Reviewer comment: Please, add references.

Response: We have added references to the manuscript.

Reviewer comment: Please, review the sum of all records identified. Based on my calculations it should be “1,862” the figure reads “Records identified n=1,857”.

Response: We have revised Figure 1 and updated the text accordingly.

Reviewer comment: Add the limitations of the study at the end of the discussion section.

Response: We have moved the limitations section to the end of the discussion section and added a section sub-header.

Reviewer 2 Report

First of all, the main message the introduction is trying to convey is not clear. The introduction does not clearly explain what, why and how the Authors are trying to study. In particular, the Authors do not clearly explain the motivation behind the choice of the research topic. Why should it be studied in the first place? Why is it important? The main goal of the paper is not clearly described. The contribution to the literature and the summary of the key findings are missing. Moreover, the paper structure has not been discussed. Therefore, I would propose to restructure the introduction according to the following more usual format:

- Overview and motivation

- Problem Statement

- Objectives of the Study

- Research Contribution

- Research Questions and the Methodology the study applies

- Key Findings of the Study

- The structure of the paper

The concluding section is too long. I would suggest to split it into two parts. The first part should contain the summary of the main findings while the second the remaining elements. In particular, the second part should discuss the limitation of the employed approach, identify the shortcomings of the existing studies and propose directions for futher research and finally provide specific recommendations for businesses and goverment policies. 

Author Response

Thank you for reviewing our paper and providing suggestions for revisions. These suggestions have helped us to strengthen the front end and conclusion of the paper.

Reviewer Comment: First of all, the main message the introduction is trying to convey is not clear. The introduction does not clearly explain what, why and how the Authors are trying to study. In particular, the Authors do not clearly explain the motivation behind the choice of the research topic. Why should it be studied in the first place? Why is it important? The main goal of the paper is not clearly described. The contribution to the literature and the summary of the key findings are missing. Moreover, the paper structure has not been discussed. Therefore, I would propose to restructure the introduction according to the following more usual format:

- Overview and motivation

- Problem Statement

- Objectives of the Study

- Research Contribution

- Research Questions and the Methodology the study applies

- Key Findings of the Study

- The structure of the paper

Response: We have revised the introduction to better articulate the main goal of the paper. We have added a third paragraph which includes a road map (structure of the paper) and a summary of the methodology, results, and key findings.

Reviewer Comment: The concluding section is too long. I would suggest to split it into two parts. The first part should contain the summary of the main findings while the second the remaining elements. In particular, the second part should discuss the limitation of the employed approach, identify the shortcomings of the existing studies and propose directions for futher research and finally provide specific recommendations for businesses and goverment policies.

Response: We have divided the discussion into two sections, with the second section focused on limitations and shortcomings of the existing study and directions for future research. We did not include specific recommendations for businesses and government policies, because the lead author is an employee of the USDA, Economic Research Service, and the agency does not permit researchers and research products authored by ERS researchers to make policy recommendations.

Reviewer 3 Report

Thank you for the opportunity to review this manuscript, a systematic review examining marketing practices for online grocery retailers and what the implications may be for WIC participants. This topic is of high importance and the study addresses an important literature gap, with there currently being little available evidence surrounding the potential impacts of online grocery purchasing methods on users of programs such as WIC.

I have outlined a few comments to help clarify several aspects of the manuscript:

Overall / Abstract:

·        Very interesting study overall. Research on the online food retail environment is lacking in academic literature, so it is good to see a robust study.

·        Abstract has too much background and too little results

·        Abstract has no concluding statement, what is the key take home message of this study?

Introduction:

·        Lines 30-35 – more explicit explanation of what WIC actually is for international readers, i.e. provides funds in the form of an EBT card etc for participants to purchase food from an approved list

·        Line 69 – options to purchase groceries online have existed for decades, citation needed

·        Line 69 – specify COVID-19 pandemic

·        Line 155 – citation for waiver of requirement for WIC to be redeemed in the presence of a clerk

Methods:

·        Line 266 – add justification for only including studies post 2015

·        Line 299 – add citation for Four Ps

Results:

·        Lines 306-322; Figure One – not clear where the additional study from the right hand side of the flow diagram has come from, is the right side your grey lit search? Results text suggests that there were no results from grey lit. Results text shows 18 studies included, but flow diagram shows 18 + 1

·        Figure one – consistency between use of “records” and “reports”

Discussion:

·        Line 522 - The third research question was to consider the implications of marketing strategies used for online groceries for WIC participants. As no studies were found that examined this specific study population, is this not a literature gap? Highlighting the need for research that specifically examines WIC and online grocery shopping would help to answer your third research question

·        Line 529 – I disagree that not including marketing content from social media, television etc is a limitation of the review as these types of marketing do not facilitate purchasing in the same way that marketing used by grocery store websites/apps

·        Line 559 – studies that examine grocery delivery may be relevant to WIC participants as they are often used by women with young children

·        Line 588 – are there previous studies among SNAP or low income participants that you could use to compare your findings?  

·        Line 622 – online nutrition labelling was found to influence healthier purchases across all included studies, but some studies have found that it has no effect: https://onlinelibrary.wiley.com/doi/abs/10.1111/j.1753-6405.2011.00684.x

·        Line 622 – additionally, other studies have shown that nutritional information is often not available when shopping online: https://www.mdpi.com/2072-6643/13/8/2611

·        No conclusion?

Author Response

Thank you for reviewing our paper and providing suggestions for revisions. These suggestions have helped us to strengthen the paper.

Reviewer Comment: Very interesting study overall. Research on the online food retail environment is lacking in academic literature, so it is good to see a robust study.

Response: Thank you for this comment on the importance of the paper topic.

Reviewer Comment: Abstract has too much background and too little results. Abstract has no concluding statement, what is the key take home message of this study?

Response:  We have revised the abstract to remove background details and include a summary of the findings and key take aways.

Reviewer comment: Lines 30-35 – more explicit explanation of what WIC actually is for international readers, i.e. provides funds in the form of an EBT card etc for participants to purchase food from an approved list

Response: We have added this additional context about WIC (line 37).

Reviewer comment: Line 69 – options to purchase groceries online have existed for decades, citation needed

Response: We have added this citation (now line 87).

Reviewer comment: Line 69 – specify COVID-19 pandemic

Response: We have specified COVID-19 pandemic (now line 87).

Reviewer comment: Line 155 – citation for waiver of requirement for WIC to be redeemed in the presence of a clerk.

Response: We have added the citation to 7 CFR § 246.12 (now line 163).

Reviewer comment: Line 266 – add justification for only including studies post 2015

Response: We have provided a justification for including studies in 2015 and later (line 266).

Reviewer comment: Line 299 – add citation for Four Ps

Response: We have added a citation to the Four Ps at line 303 and in the introduction at line 78.

Reviewer comment: Lines 306-322; Figure One – not clear where the additional study from the right-hand side of the flow diagram has come from, is the right side your grey lit search? Results text suggests that there were no results from grey lit. Results text shows 18 studies included, but flow diagram shows 18 + 1. Figure one – consistency between use of “records” and “reports”

 Response: We have revised Figure 1 and the corresponding text to include the correct number of included studies and to replace “reports” with “records”.

Reviewer comment: Line 522 - The third research question was to consider the implications of marketing strategies used for online groceries for WIC participants. As no studies were found that examined this specific study population, is this not a literature gap? Highlighting the need for research that specifically examines WIC and online grocery shopping would help to answer your third research question

Response: We have added the need for future studies focused on WIC participants/populations served by WIC at line 591.

Reviewer comment: Line 529 – I disagree that not including marketing content from social media, television etc. is a limitation of the review as these types of marketing do not facilitate purchasing in the same way that marketing used by grocery store websites/apps

Response: We have noted that this is not a limitation, per se, at line 612.

Reviewer comment: Line 559 – studies that examine grocery delivery may be relevant to WIC participants as they are often used by women with young children

Response: We have added this consideration at line 624.

Reviewer comment: Line 588 – are there previous studies among SNAP or low income participants that you could use to compare your findings?  

Response: No studies that we identified in this review focused on participants of other means-test or nutrition assistance programs. One study conducted subgroup analyses among low-income individuals, which we note at line 498.

Reviewer comment: Line 622 – online nutrition labelling was found to influence healthier purchases across all included studies, but some studies have found that it has no effect: https://onlinelibrary.wiley.com/doi/abs/10.1111/j.1753-6405.2011.00684.x

Response: We have incorporated these findings at line 571. 

Reviewer comment: Line 622 – additionally, other studies have shown that nutritional information is often not available when shopping online: https://www.mdpi.com/2072-6643/13/8/2611

Response: We have added this citation at line 578.

Reviewer comment: No conclusion?

Response: We have added a conclusion at line 629.